# Bioactivity of Fractions and Pure Compounds from *Jatropha cordata* (Ortega) Müll. Arg. Bark Extracts

**DOI:** 10.3390/plants12213780

**Published:** 2023-11-06

**Authors:** Yazmín B. Jiménez-Nevárez, Julio Montes-Avila, Miguel Angel Angulo-Escalante, Ninfa Yaret Nolasco-Quintana, Judith González Christen, Israel Hurtado-Díaz, Eber Addí Quintana-Obregón, J. Basilio Heredia, José Benigno Valdez-Torres, Laura Alvarez

**Affiliations:** 1Centro de Investigación en Alimentación y Desarrollo, A.C. Carretera Eldorado km 5.5, Campo El Diez, Culiacán 80110, Mexico; yasmin.jimenez.dc18@estudiantes.ciad.mx (Y.B.J.-N.); mangulo@ciad.mx (M.A.A.-E.); jbheredia@ciad.mx (J.B.H.); 2Programa de Posgrado en Ciencias Biomédicas, Facultad de Ciencias Químico-Biológicas, Universidad Autónoma de Sinaloa, Ciudad Universitaria s/n, Culiacán 80010, Mexico; jmontes@uas.edu.mx; 3Centro de Investigaciones Químicas, Instituto de Investigación en Ciencias Básicas y Aplicadas, Universidad Autónoma del Estado de Morelos, Avenida Universidad 1001, Col. Chamilpa, Cuernavaca 62209, Mexico; ninfa.nolascoqui@uaem.edu.mx; 4Laboratorio de Inmunidad Innata, Facultad de Farmacia, Universidad Autónoma del Estado de Morelos, Avenida Universidad 1001, Col. Chamilpa, Cuernavaca 62206, Mexico; judith.gonzalez@uaem.mx; 5Departamento de Madera Celulosa y Papel, Centro Universitario de Ciencias Exactas e Ingenierías, Universidad de Guadalajara, km 15.5 Guadalajara-Nogales, Las Agujas, Zapopan 45100, Mexico; ihurtado@uaem.mx; 6CONACYT—Centro de Investigación en Alimentación y Desarrollo, A.C. Carretera Gustavo Enrique Astiazarán Rosas, No. 46, Col. La Victoria, Hermosillo 83304, Mexico; eber.quintana@ciad.mx

**Keywords:** *Jatropha cordata* (Ortega) Müll. Arg., anti-inflammatory activity, NO inhibition, campesteryl palmitate, n-heptyl ferulate, palmitic acid, mixture of sterols

## Abstract

Medicines for chronic inflammation can cause gastric ulcers and hepatic and renal issues. An alternative treatment for chronic inflammation is that of natural bioactive compounds, which present low side effects. Extracts of *Jatropha cordata* (Ortega) Müll. Arg. have been evaluated for their cytotoxicity and anti-inflammatory activity; however, testing pure compounds would be of greater interest. Campesteryl palmitate, n-heptyl ferulate, palmitic acid, and a mixture of sterols, i.e., brassicasterol, campesterol, β-sitosterol, and stigmasterol, were obtained from an ethyl acetate extract from *J. cordata* (Ortega) Müll. Arg. bark using column chromatography. The toxicity and in vitro anti-inflammatory activities were evaluated using RAW 264.7 murine macrophage cells. None of the products assessed exhibited toxicity. The sterol mixture exhibited greater anti-inflammatory activity than the positive control, and nitric oxide (NO) inhibition percentages were 37.97% and 41.68% at 22.5 μg/mL and 30 μg/mL, respectively. In addition, n-heptyl ferulate decreased NO by 30.61% at 30 μg/mL, while campesteryl palmitate did not show anti-inflammatory activity greater than the positive control. The mixture and n-heptyl ferulate showed NO inhibition; hence, we may conclude that these compounds have anti-inflammatory potential. Additionally, further research and clinical trials are needed to fully explore the therapeutic potential of these bioactive compounds and their efficacy in treating chronic inflammation.

## 1. Introduction

Certain forms of cancer, cardiovascular and cerebrovascular diseases, diabetes, and several other diseases have been strongly linked to chronic inflammatory processes [1,2,3,4]. Until now, the main treatments for these processes include the use of synthetic medicines, whose prolonged consumption may lead to undesirable side effects such as gastric ulcers, and liver, kidney, heart, and other problems [5]. In the search for alternatives to synthetic drugs, natural products with minor secondary effects have received significant attention for the treatment of chronic inflammation [6,7]. However, bioaccessibility, cellular cytotoxicity, and other aspects of these bioproducts are still an important concern regarding their use as a regular therapy. Hence, obtaining and identifying new chemical constituents responsible for anti-inflammatory activity is an active area of research that may lead to more effective and secure bioactive compounds for the treatment of several health conditions and chronic inflammatory processes.

Trees from several species of the genus *Jatropha*, from the *Euphorbiaceae* family, have received special attention in the search for compounds with diverse bioactivities as several parts are traditionally used as treatments for injuries and inflammatory processes associated with gastrointestinal problems in India, México, and other countries [8,9,10]. These uses prompted research focused on studying diverse extracts from the leaf, fruit, latex, root, and bark from several species of Jatropha. It was found that several species of *jatropha* are rich in bioactive compounds, such as cyclic peptides, lignans [11], flavonoids [12], coumarins [13], alkaloids [14], monoterpenes [15], sesquiterpenes [16], diterpenes [16], and triterpenes [17]. These compounds exhibit several biological activities that make them excellent candidates for the design of products that might contribute to the treatment of diverse diseases.

Well adapted to extreme environmental conditions, 48 species of *Jatropha* have been identified in several regions of México. *Jatropha curcas*, *cinerea*, and *platyphylla*, among others, have been recently studied in Sinaloa, a northwestern state of the country [10,18]. *Jatropha cordata* (Ortega) Müll. Arg. is an endemic plant from México, whose bark is commonly used by the population as an anti-inflammatory treatment with positive results, although it has received limited attention as a source of bioactive compounds. Hence, our research aimed to obtain, identify, and assess bioactive compounds from the bark of *J. cordata* (Ortega) Müll. Arg. In this stage of our research, a bio-directed technique was designed to generate pure compounds or mixtures with anti-inflammatory action, utilizing an ethyl acetate extract derived from our previous work [10]. The process consisted of the fractionation, purification, identification, and quantification of pure compounds and mixtures obtained from this extract to evaluate their cellular cytotoxicity and their ability to inhibit NO production in an in vitro model of RAW 264.7 murine macrophages.

## 2. Results

### 2.1. Fractionation Yield

Beginning with 16.6 g of crude ethyl acetate extract, two fractions, labelled F-1 and F-2, were obtained by column chromatography, with yields of 2.99 g and 5 g, respectively. Fraction F-2 showed the best NO inhibition effect. By means of silica gel column chromatography, two subfractions were produced, labelled SF-1 and SF-2, with yields of 1.42 g and 3.21 g, respectively. From SF-1, the compound campesteryl palmitate was obtained by column chromatography, with a yield of 3.6 mg. Chromatographic separation of SF-2 allowed the isolation of n-heptyl ferulate and a mixture of four sterols with yields of 7 mg and 148.5 mg. The mixture components were identified as brassicasterol, campesterol, β-sitosterol, and stigmasterol using spectroscopic characteristics and a comparison of the spectral data with those in the literature. A GC-MS analysis showed that the percentages of brassicasterol, campesterol, β-sitosterol, and stigmasterol in the mixture were 8.74%, 22.8%, 18.30%, and 29.20%, respectively.

### 2.2. NMR

Campesteryl palmitate was identified by the characteristic signals in its NMR spectra, with ^1^H NMR displaying a one-proton broad singlet at δ_H_ 5.30 (brs) assigned to the olefinic proton H-6, one hydroxyl proton (δ_H_ 4.53, td), and seven methyl protons, six ascribable to the sterol nucleus [δ 0. 82, (3H, s, H-18), 0.81 (3H, d, *J* = 6.4 Hz, H-21), 0.80 (3H, d, *J* = 6.8 Hz, H-27), 0.78 (3H, d, *J* = 6.4 Hz, H-26), 0.62 (3H, d, *J* = 7 Hz, H-28), and 0.60 (3H, s, H-19)], and one triplet at δ_H_ 0.814 corresponding to the fatty acid chain. Also, at δ_H_ 2.20, a two-proton triplet typical of the methylene attached to the carbonyl ester was observed, and the single signal integrated for 64 protons was assigned to the fatty ester chain. Its ^13^C and DEPT spectra displayed six methyl carbons (21.25, 20.04, 14, 34 (X2), 12.20, and 12.08), nine methylene carbons (122.80, 77.38, 56.92, 56.26, 50.25, 38.39, 36.38, 34.17, and 32.09), ten methylene carbons (42.54, 39.95, 37.23, 34.96, 34.17, 32.09, 31.20, 26.29, 24.52, and 23.29), and three quaternary carbons (139.96, 46.06, and 36.83), which matched with those described for campesterol [19]. In addition, the signal at δ 173.57 was attributed to a carboxyl group, together with the methylene signals at δ_C_ 32.09, 31.20, 29.87, 29.81, 29.67, 29.58, 29.48, and 29.37, and the methyl signal at δ_C_ 19.55, which were assigned to the palmitic acid chain. This evidence allowed the identification of this compound as campesteryl palmitate (Figure 1). Although this compound has been described as a constituent of many plant species, this is the first report of its complete NMR spectral data. Figure A1 and Figure A2, in Appendix A, correspond to the ^1^H-NMR spectrum of campesteryl palmitate and the ^13^C-NMR spectrum of campesteryl palmitate, respectively.

In addition, n-heptyl ferulate was identified by comparing its NMR data with those described in [20]. Its ^13^C NMR spectrum, and the HSQC spectrum, led to the identification of 17 carbon resonances, which were assigned to two CH_3_, six CH_2_, five CH, and four quaternary carbon atoms. Among the carbon resonances, the signals at δ_C_ 167.5 (C-9), 144.8 (C-7), and 115.8 (C-8) were assigned to the α, β-unsaturated carboxyl; and the six aromatic carbon signals at δ_C_ 127.2 (C-1), 109.39 (CH-2), 146.8 (C-3), 148.0 (C-4), 114.9 (CH-5), and 123.1 (CH-6) correspond to the trisubstituted benzene ring; the signal at δ_C_ 56.9 is ascribable to a methoxyl group. Also, this spectrum reveals the presence of a C7 saturated hydrocarbon chain, evidenced by the seven signals at δ_C_ 64.70 (C-1′), 32.0 (C-6′), 29.6 (C-2′), 29.4 (C-4′), 28.8 (C-3′), 22.7 (C-5′), and 14.2 (C-7′). The ^1^H-NMR spectrum reveals signals for an AB olefinic spin system at δ_H_ 7.59 (d, *J* = 15.9 Hz, H-7) and 6.27 (d, *J* = 15.9 Hz, H-8), and one aromatic ABX spin system at δ_H_ 7. 06 (dd, *J* = 8.1, 1.8 Hz, H-6), 7.02 (d, *J* = 1.8 Hz, H-2), and 6.9 (d, *J* = 8.1 Hz, H-5), indicating a 1,3,4 trisubstituted benzene ring. In addition, signals for a methoxyl group (δ_H_ 3.91, s) and one OH group (δ_H_ 5.85, sbr) were also observed. The aliphatic side chain was evidenced by the signals at δ_H_ 4.17 (2H, t, *J* = 6.7 Hz, H-1′), 1.67 (2H, q, H-2′), 1.36 (8H, m, H-3′-H-6′), and 0.86 (3H, t, *J* = 6.8 Hz, H-7′). Furthermore, these signals were verified by proton–carbon correlations in the HMBC spectrum. Therefore, compound 2 was determined as n-hepthyl ferulate based on the above-mentioned experimental results and previous reports from the literature [20] (Figure 1). Experimentally, n-heptyl ferulate was obtained as yellow crystals, and its molecular weight was determined by the positive FAB/MS ion peak at *m*/*z* = 315 [M + Na]^+^. Figure A3, Figure A4, Figure A5 and Figure A6, in Appendix A, correspond to the spectra ^1^H-NMR, ^13^C-NMR, HSQC, and HMBC of n-heptyl ferulate, respectively.

The mixture showed NMR spectra data highly similar to brassicasterol, campesterol, β-sitosterol, and stigmasterol as reported in the literature, with ^1^H NMR displaying δ_H_ for one olefinic methine proton at δ_H_5. 27, brd, ascribable to H-6 of the steroid nucleus, and two more methine protons at δ_H_ 5.08 (dd, *J* = 15.1, 8.7 Hz) and 4.94 (dd, *J* = 15.1, 8.8 Hz) corresponding to the double bond at C-22/C-23 of the side chain of brassicasterol and stigmasterol. The hydroxyl proton H-3 appeared at δ_H_ 3.44 as only one signal for the four sterols. The ^13^C NMR and DEPT revealed the presence of nine methyl carbons (δ_C_ 20.02, 19.60, 19.24, 19.19, 18.98, 12.45, 12.25, 12.19, and 12.06); eighteen methylene carbons (δ_C_ 42.52, 42.48, 42.41, 39.97, 39.82, 37.45, 34.09, 32.16, 31.79, 29.92, 29.11, 28.46, 26.24, 25.60, 24.60, 24.48, 23.26, and 21. 29); twelve aliphatic methine carbons (δ_C_ 57.07, 56.97, 56.26, 56.16, 51.44, 5034, 46.03, 40.72, 36.71, 32.16, 29.36, and 21.42), one methine carbinol at δ_C_ 71. 94 and three olefinic methines at δ_C_ 138.45 (C-22), 129.45 (C-23), and 121.82 (C-6); and three quaternary carbons at δ_C_ 141.91, 46.03, and 36.30 (Figure 1).

The four steroids present in this mixture have the same steroidal skeleton and only differ in the side chain, i.e., through the presence of a methyl group for campesterol and an ethyl group for β-sitosterol in C-24, and through the presence of a double bond at C-22 for brassicasterol and at C-23 for stigmasterol in the side chain. By comparing the NMR data of the mixture with those described in the literature for the individual compounds, and the C-H correlations of the HSQC and HMBC spectra, it was possible to identify the signals corresponding to each one, as is described in the experimental section (Figure 1). Figure A7 and Figure A8, in Appendix A, correspond to the ^1^H-NMR spectrum of the mixture of sterols (brassicasterol, campesterol, β-sitosterol, and stigmasterol) and the ^13^C-NMR spectrum of sterols (brassicasterol, campesterol, β-sitosterol, and stigmasterol).

### 2.3. In Vitro Anti-Inflammatory Activity

#### 2.3.1. Cell Cytotoxicity

The cytotoxicity of fractions F-1 and F-2 at concentrations of 3.125, 6.25, 12.5, 25, and 50 μg/m was assessed for their effect on viability in murine RAW 264.7 macrophage cells. Lipopolysaccharides (LPS) and lipopolysaccharides + indomethacin (LPS + Indo) were used as negative (C−) and positive (C+) controls, respectively. The corresponding analysis of variance (Table 1) showed that concentration was the only significant factor, and all concentration levels were equal and significantly different from the negative control (Table 2); hence, both fractions did not show cell cytotoxicity at the concentrations used in the experiment (Figure 2).

Similar results occurred for subfractions SF-1 and SF-2, at 3. 75, 7.5, 15, 22.5, and 30 μg/mL, as shown in Table 3 and Table 4 and Figure 3. No subfractions at any concentrations resulted in cytotoxic in murine RAW 264.7 macrophage cells.

The analysis of variance showed that the compounds (campesteryl palmitate, C-1, n-heptyl ferulate, C-2, and the steroid mixture) and concentration were statistically significant (Table 5). Tukey’s test found C-2 to be different from C-1 and the mixture (Table 6). Neither concentration resulted in cell cytotoxic compared to the negative control (C−) (Table 7). Figure 4 shows profiles for the two compounds and the mixture with respect to concentration and confirms the above interpretation. The results showed no macrophage toxicity of compounds or the mixture at any of the studied concentrations.

#### 2.3.2. Nitric Oxide (NO) Inhibition

The potential anti-inflammatory activity of fractions, subfractions, and compounds was assessed using the effects on NO inhibition in murine RAW 264.7 macrophage cells stimulated with LPS.

The inhibition of NO from fractions F-1 and F-2 was studied at concentrations of 3.125, 6.25, 12.5, 25, and 50 μg/mL. The analysis of variance (Table 8) shows that concentration was the only significant factor (*p*-value < 0.05). Tukey’s test showed that none of the concentrations were statistically different of C+, and all were better than C− (Table 9). Figure 5 shows that only F-2 at a concentration of 50 μg/mL was better than C+.

The effects of subfractions and their concentrations on NO inhibition are shown in the following. The analysis of variance (Table 10) shows that only the concentration was statistically significant (*p* < 0.05). Tukey’s test (Table 11) found that concentrations from 15 to 30 μg/mL had comparable effects to C+. In fact, Figure 6 shows that SF-1 and SF-2 were comparable to C+ at 30 μg/mL.

Finally, in the case of compounds, both factors were statistically significant (*p* < 0.05) (Table 12). According to Tukey’s test, campesteryl palmitate and the steroid mixture exhibited greater NO inhibition than n-heptyl ferulate (Table 13 and Table 14). However, only the steroid mixture at concentrations of 22.5 and 30 μg/mL produced better results than C+ (Figure 7).

## 3. Discussion

The anti-inflammatory ethyl acetate extract of *Jatropha cordata* (Ortega) Müll. Arg. was separated using the NO inhibition bioassay, and an active SF-2 was obtained. This fraction was mainly composed of steroids (70%). SF-2 was further purified using column chromatography to yield campesteryl palmitate, n-heptyl ferulate, and a mixture of four sterols, identified as brassicasterol, campesterol, β-sitosterol, and stigmasterol (Figure 1) based on their spectroscopic characteristics and a comparison of the spectroscopic data with those in the literature. A quantitative analysis showed that the percentages of brassicasterol, campesterol, β-sitosterol, and stigmasterol in the SF were 8.74%, 22.8%, 18.30%, and 29.20%, respectively.

Plant sterols are naturally occurring bioactive compounds in plant materials [21,22]. They are highly present in lipid-rich plant foods such as nuts, seeds, legumes, and olive oil and have been shown to elicit a broad range of pharmacological activities, such as antiallergy, antitumor [23], antimalarial, antiobesity, antimicrobial [24], antidepressant [25], antinociceptive [26], and antileishmanial activities [27], cardiovascular protection [28], and antiaging and hepatoprotective activities [29]. All plant species have their characteristic phytosterol (PS) composition, with more than 250 PS being recognized so far [30]. Campesterol, β-sitosterol, and stigmasterol are the most common plant-derived sterols in the human diet. All contain a core skeleton of cholesterol but possess a different side chain. β-sitosterol and stigmasterol have an ethyl group at C-24, whereas campesterol has a methyl group. Stigmasterol has a double bond at C-22/C-23, and sitosterol has a saturated side chain. The compounds brassicasterol and D-7-avenasterol are minor constituents [31].

The compound campesteryl palmitate did not inhibit NO production at the concentrations evaluated, although this compound has been reported as a constituent of a wide variety of plants and nutraceuticals [32]. Reports of its biological activities were not discovered during our research. Furthermore, Compound **1** did not inhibit NO production at the concentrations evaluated compared to fractions and subfractions of *J. cordata* (Ortega) Müll. Arg., which showed significant activity.

The compound n-heptyl ferulate displayed an important anti-inflammatory effect since it diminished the concentration of NO by 30.61% at 30 μg/mL. This compound is an ester derivative of ferulic acid, which is a hydroxycinnamic acid widely distributed in cereals, fruits, vegetables, and beverages [33,34]. In these foods, Compound **2** is found in its free form and as ester derivatives, displaying a wide range of biological activities, including anticancer, antibacterial, anticarcinogenic, and anti-inflammatory activity [35,36]. Esterification has shown some advantages over precursor compounds [37,38,39,40]. In this context, the compound n-heptyl ferulate has been previously reported as a natural product from *Jatropha podagrica* [20], but no activity was assessed. This is the first report to focus on the anti-inflammatory activity of n-heptyl ferulate.

Another pure compound isolated and identified was palmitic acid, which has already been widely reported in the literature with pharmacological activities such as antiviral [41,42], anti-inflammatory [41,43], analgesic [41,44], and lipid metabolism-regulating activities [41,45]. Regarding its anticancer activity, several authors have reported that palmitic acid induces cell cycle arrest [41,46] and promotes apoptosis of human neuroblastoma cells [41,47] and breast cancer cells [41,48]. In addition, palmitic acid can inhibit hepatoma cell proliferation by changing membrane fluidity and blocking glucose metabolism [41,49]. Cantrell et al. [50] demonstrated that palmitic acid obtained from *Jatropha curcas* L. at a concentration of 25 nmol/cm^2^ acts as a repellent against *Aedes aegypti* (L.) mosquitoes (*Diptera*: *Culicidae*). Othman et al. [51] reported that palmitic acid obtained from fractions of n-hexane extracts of *J. curcas* root presented anti-inflammatory activity in RAW 264.7 cells at an effective concentration of 1 mg/mL. Aati et al. [8], studying palmitic acid obtained from *J. pelargoniifolia* root essential oil, reported anti-inflammatory, antipyretic, anticonceptive, analgesic, and antioxidant activity. Regarding antimicrobial activity, Shaaban et al. [44] reported that palmitic acid has an effect against *S. aureus*, *P. aeruginosa*, *K. pneumoniae*, *Lalthanpuii*, and *Lalchhandama*. Shaaban et al. [52] mentioned that the antimicrobial effect of palmitic acid is due to its structure, shape, the length of its carbon chains, and the presence, number, position, and orientation of double bonds. As for its anticancer activity, Zhu et al. [41] tested palmitic acid in human prostate cancer cell lines PC3 and DU145 at 0.1, 1, 1, 5, 10, 25, and 50 μM concentrations, concluding that it has anticancer activity in prostate cancer by arresting G1 phase and suppressing tumor metastasis regulatory proteins. The underlying mechanism of these effects could be attributed to the inhibition of the PI3K/Akt pathway. Diverse biological activities of palmitic acid have been reported by [53,54] who, beginning with a methanolic extract of *Chrozophora tinctoria* (L.) from the family *Euphorbiaceae*, at concentrations of 1000, 500, 250, and 125 µg/mL, reported antioxidant, nematicidal, pesticidal, hypocholesterolemic, nematicidal, hemolytic, and 5-alpha reductase inhibitory activities.

The compounds that constitute the mixture of free and esterified sterols are brassicasterol, campesterol, β-sitosterol, and stigmasterol. Brassicasterol has been reported for its antiaging activity under oxidative stress and decreased ROS and MDA levels [55] and against Alzheimer’s disease, which is attributed to its bioactivity against amyloid beta and tau receptors [56]. Similarly, antiherpes simplex type I and antituberculosis activities attributed to the inhibition of vital enzymes involved in HSV-1 replication and Mtb cell wall biosynthesis have been reported [57], as have the inhibitory properties of human angiotensin-converting enzyme [58]. Kuwabara et al. [59] reported the accumulation of cholesterol precursors (latosterol, 7-dehydrocholesterol, and desmosterol) and their decrease by altering mRNA and biosynthesis protein levels, increasing sterol 8,7-isomerase (EBP) enzymes, and decreasing DHCR7 and 24-dehydrocholesterol reductase (DHCR24).

Brassicasterol has been reported for its anticancer activity in prostate cancer, which was attributed to AKT and AR dual-targeting signaling [60], and in bladder cancer through its androgen receptor (AR) antagonist action and AR (androgen receptor) expression-reducing effect in bladder epithelial cells [61]. Brassicasterol has also demonstrated activity in hepatocellular carcinoma [58] through the suppression of the AKT signaling pathway.

As for the compound stigmasterol, Viswanatham et al. [62] reported antimicrobial activity in Gram-positive bacteria such as *Bacillus cereus*, *Bacillus subtilis*, *Staphylococcus aureus*, and *Staphylococcus epidermis*; in Gram-negative bacteria such as *Aeromonas hydrophila*, *Escherichia coli*, *Klebsiella pneumoniae*, *Pseudomonas aeruginosa*, *Proteus mirabilis*, *Proteus vulgaris*, *Salmonella paratyphi*, *Salmonella paratyphi A*, *Vibrio alcaligenes*, and *Vibrio cholerae*; and antifungal activity in *Aspergillus fumigatus*, *Candida albicans*, *Microsporum gypseum*, and *Trichophyton rubrum* fungi at concentrations of 50, 25, and 12. 5 mg/mL for bacteria, and at 10, 5, and 2.5 mg for fungi, respectively. Finally, β-sitosterol has been reported as having antioxidant, anticancerogenic, larvicidal (mosquitoes), and antimicrobial activity [63,64]. On the other hand, in [65], methanol extracts from *J. curcas* seeds were shown to contain β-sitosterol (13% *w*/*w*) using GC-MS, which exhibited antimicrobial activity against Gram-positive and Gram-negative pathogenic bacteria (inhibition range: 0–1.63 cm) at concentrations of 1 and 1.5 mg/disc.

In addition, mixtures of the above compounds have been shown to have diverse biological activities. Dumandan et al. [66] showed that the mixture of brassicasterol, campesterol, and stigmasterol presented antimicrobial activity. Prabhakar et al. [67] found that the mixture of brassicasterol, campesterol, and β-sitosterol presented activity against androgenic alopecia as they were potential inhibitors of 5α-reductase1. Abou-Hussein et al. [68] determined that the mixture of brassicasterol and campesterol manifested anti-inflammatory activity in vivo. Akintayo [69] and Sekandí et al. [70] reported that a mixture of campesterol, β-sitosterol, and stigmasterol exhibited antioxidant, antimicrobial, and sunscreen activities. Hérnandez-Hérnandez et al. [71] reported that the mixture of β-sitosterol and stigmasterol had antioxidant, antimicrobial, and antifungal activities. Finally, Mahrous et al. [72] found anti-inflammatory activity, attributing it to the decrease in NO, prostaglandin PGE2, TNF-α, and PKC levels by 19, 29.35, 16.9, and 47.83%, respectively.

## 4. Materials and Methods

### 4.1. Materials and Reagents

The research material for this study was an ethyl acetate extract of *Jatropha cordata* (Ortega) Müll. Arg. bark obtained previously [10]. Silica gel (70–230 mesh, ASTM, and 230–400 mesh) used for preparative thin-layer chromatography (TLC) was purchased from Merck. Open column chromatographies were carried out on silica gel 60 (70–230 mesh), and different solvent systems of n-hexane and EtOAc were used as mobile phases for purification. Deuterated chloroform (CDCl_3_), indomethacin, LPS from *Escherichia coli* serotype 055:B5, sodium nitrite (NaNO_2_), *N*-(1-naphthyl)ethylenediamine dihydrochloride, and sulfanilamide were purchased from Sigma Aldrich (St. Louis, MO, USA). Dulbecco’s modified Eagle’s medium/nutrient mixture F-12 (DMEM/F12), fetal bovine serum (FBS), and glutamine (GlutaMax) were purchased from GIBCO (New York, NY, USA), and [3-(4,5-dimethyl-2-yl)-5-(3-carboxymethoxyphenyl)-2-(4-sulfophenyl)-2*H*-tetrazolium, inner salt; MTS] was purchased from Promega Co. (Madison, WI, USA). NMR spectra were recorded at 600 MHz for ^1^H-NMR and 150 MHz for ^13^C-NMR in the presence of tetramethylsilane (TMS) as the internal standard in CDCl_3_ on a Jeol 600 instrument, (Tokyo, Japan) and at 500 MHz for ^1^H and 100 MHz for ^13^C on a Bruker 500 instrument as indicated. Chemical shifts δ are expressed in parts per million (ppm) relative to TMS, and coupling constants (*J*) in Hertz. Multiplicities are indicated as singlet (s), doublet (d), triplet (t), quartet (q), double of double (dd), multiplet (m), and broad singlet (bs). Mass spectra were obtained in a Jeol M-station JEOL JMX-AX 505 HA mass spectrometer.

### 4.2. Fractionation

The ethyl acetate extract was obtained from 100 g of *J. cordata* (Ortega) Müll. Arg. bark, with a yield of 16.6 g. The extract was subjected to normal phase column chromatography (CC) on 350 g of silica gel (63–200 mesh) and gradient elution with *n*-hexane and AcOEt. Sixty-seven 200 mL fractions were collected and pooled into two fractions according to their similarity in CCF: F-1 (1–24, 3.2 L, 2.99 g, 100:0–0:20); F-2 (25–54, 6 L, 5.21 g, 75:25–30:70), and 5 g of F-2 was subjected to column chromatography (CC) on 120 g of silica gel and elution with a hexane:acetone gradient. Fractions 2–21 (1.4 g) eluted with 95:5–93:7 (1L) were pooled into the SF-1 subfraction, while fractions 22–39 (3.21 g) eluted with 91:9–85:15 (1.8 L) were pooled into the SF-2 subfraction.

SF-1 (1.4 g) was separated into its components using silica gel column chromatography (52 g) and dichloromethane: MeOH gradient elution, yielding 6 fractions of 50 mL. Fractions 4–6, eluted with DCM: MeOH (98:2) (161 mg), were subsequently rechromatographed on DC (8.5 g silica gel), eluting with *n*-hexane-AcOEt (92:8), to give 31.6 mg, which was purified using preparative thin layer chromatography (TLCP) with hexane: DCM (7:3) (6 developments) to give 3.6 mg of campesteryl palmitate.

SF-2 (2.18 g) was rechromatographed on a silica gel column (100 g), eluting with CH_2_Cl_2_. Thirty-two 50 mL fractions were obtained. Fraction 9 contained a pure compound (7 mg) characterized as n-heptyl ferulate. Fractions 17–21 contained a white precipitate (148.5 mg), which was filtered and characterized as the mixture of four sterols identified as brassicasterol, campesterol, β-sitosterol, and stigmasterol. The mixture of 3–6 was subsequently acetylated with Ac_2_O/py at room temperature for 12 h, and the product was analyzed using GC-MS to determine the relative concentration of the components in the mixture.

### 4.3. NMR

Campesteryl palmitate: waxy solid, ^1^H-NMR (500 MHz, CDCl_3_), δ: 5.30 (1H, m, H-6), 4.53 (1H, td, H-3), 2.20 (2H, m, H-1′), 1.17 (64H, m), 0.82, (3H, s, H-18), 0.81 (3H, d, *J* = 6.4 Hz, H-21), 0.80 (3H, d, *J* = 6.8 Hz, H-27), 0.78 (3H, d, *J* = 6.4 Hz), 0.62 (3H, d, *J* = 7 Hz, H-28), 0.60 (3H, s, H-19). ^13^C-NMR (100 MHz, CDCl_3_) δ: campesteryl moiety: 139.96 (C-5), 122.80 (C-6), 77.38 (C-3), 56.92 (C-14), 56.26 (C-17), 50.25 (C-9), 46.06 (C-13), 42.54 (C-4), 39.95 (C-12), 38.39 (C-24), 37.23 (C-1), 36.83 (C-10), 36.38 (C-20), 34.96 (C-22), 34.17 (C-23, C-25), 32.09 (C-8, C-7), 31.20(C-2), 26.29 (C-16), 24.52 (C-15), 23.29 (C-11), 21.25 (C-26), 20.04 (C-27), 14.34 (C-21, C-28), 12.20 (C-18), 12.08 (C-19); palmitate moiety: 173.57 (C=O), 32.09, 31.20, 29.87, 29.81, 29.67, 29.58, 29.48, 29.37, 19.55 (CH_3_).

n-Heptyl ferulate: white solid, FABMS *m*/*z* = 315 [M + Na] ^+^. ^1^H-NMR (600 MHz, CDCl_3_), δ: 7.59 (1H, d, *J*= 15.9 Hz, H-7), 7.06 (1H, dd, *J* = 8.1, 1.8 Hz, H-6), 7.02 (1H, d, *J* = 1.8 Hz, H-2), 6.9 (1H, d, *J* = 8.1 Hz, H-5), 6.27 (1H, d, *J* = 15.9 Hz, H-8), 5.85 (1H, OH), 3.91 (3H, OCH_3_), 4.17 (2H, t, *J* = 6.7 Hz, H-1′), 1.67 (2H, m, H-2′), 1.25–1.36 (8H, m, H-3′, H-4′, H-5′, H-6′), 0.86 (3H, t, *J* = 6.8 Hz, H-7′). ^13^C NMR (150 MHz, CDCl_3_) δ: 167.5 (C-9), 148.0 (C-4), 146.8 (C-3), 144.8 (C-7), 127.2 (C-1), 123.1 (C-6), 115.8 (C-8), 114.9 (C-5), 109.39 (C-2), 64.70 (C-1′), 56.9 (OCH_3_), 32.0 (C-6′), 29.6 (C-2′), 29.4 (C-4′), 28.8 (C-3′), 22.7 (C-5′), 29.38 (C-4′), 14.2 (C-7′).

Mixture of sterols brassicasterol (4), campesterol (5), β-sitosterol (6), and stigmasterol (7). ^1^H_NMR (500 MHz, CDCl_3_) δ:5.27 brs (H-6), 5.08 (dd, *J* = 15.1, 8.7 Hz, H-22), 4.94 (dd, *J* = 15.1, 8.8 Hz, H-23), 3.44 (dt, H-3); methyl protons, d_H_: 0.93, 0.85, 0.775, 0.765, 0.758, 0.750, 0.736, 0.739, 0.609. ^13^C NMR (100 MHz, CDCl_3_) δ_C_: 140.91 (C, C-5 for brassicasterol, campesterol, β-sitosterol, and stigmasterol), 138.45 (CH, C-22 for brassicasterol and stigmasterol), 129.45 (CH, C-23 for brassicasterol and stigmasterol), 121.82 (CH, C-6 for brassicasterol, campesterol, β-sitosterol, and stigmasterol), 74.1 (CH, C-3 for brassicasterol, campesterol, β-sitosterol, and stigmasterol), 57.07 (CH, C-17 for brassicasterol and campesterol), 56.97 (CH, C-17 for β-sitosterol and stigmasterol), 56.26 (CH, C-14 for β-sitosterol and stigmasterol), 56.16 (CH, C-14 for brassicasterol and campesterol), 51.44 (CH, C-9 for brassicasterol and campesterol), 50.34 (CH, C-9 for β-sitosterol and stigmasterol), 46.03 (CH, C-13, C-24 for campesterol), 42.52 (CH, C-13, C-24 for brassicasterol), 42.48 (CH_2_), 42.41 (CH_2_), 40.72 (CH, C-20 for brassicasterol and stigmasterol), 39.97 (CH_2_), 39.82 (CH, C-24 for brassicasterol), 37.45 (CH_2_), 36.71 (CH, C-20 for campesterol and β-sitosterol), 36.30 (C), 34.09 (CH_2_), 32.16 (CH, C-25 for brassicasterol and stigmasterol), 31.79 (CH_2_), 29.92 (CH_2_), 29.36 (CH, C-25 for campesterol and β-sitosterol), 29.11 (CH_2_), 28.46 (CH_2_), 26.24 (CH_2_), 25.60 (CH_2_), 24.60 (CH_2_), 24.48 (CH_2_), 23.26 (CH_2_), 21.42(CH, C-26, C-27), 21.29 (CH_2_), 20.02 (CH_3_, C-26 for stigmasterol), 19.60 (CH_3_, C-26 for β-sitosterol), 19.24 (CH_3_, C-28 for brassicasterol), 19.19 (CH_3_, C-21 for brassicasterol and campesterol), 18.98 (CH_3_, C-21 for β-sitosterol and stigmasterol), 12.45 (CH_3_, C-18 for brassicasterol), 12.25 (CH_3_, C-19 for campesterol), 12.19 (CH_3_, C-29 for β-sitosterol), 12.06 (CH_3_, C-29 for stigmasterol).

### 4.4. GC-MS Analysis

The components present in the acetylated steroidal mixture were analyzed with GC-MS using an HP Agilent Technologies 6890 gas chromatograph equipped with an MSD 5973 quadrupole mass detector (HP Agilent, Wilmington, DE, USA) and an HP-5MS capillary column (length: 30 m; inner diameter: 0.25 mm; film thickness: 0.25 M). A constant flow of carrier helium was adjusted to the column at 1 mL per minute. The inlet temperature was set at 250 °C, while the oven temperature was initially maintained at 40 °C for 1 min and increased to 280 °C at 10 °C/min intervals. The mass spectrometer started in positive electron impact mode with an ionization energy of 70 eV. Detection was performed in selective ion monitoring (SIM) mode, and peaks were identified and quantified using target ions. Compounds were identified by comparing their mass spectra with the NIST library version 1.7a. Relative percentages were determined by integrating the peaks using the GC ChemStation software, version C.00.01. The composition was reported as a percentage of the total peak area.

### 4.5. In Vitro Anti-Inflammatory Activity

The in vitro anti-inflammatory evaluation of *J. cordata* (Ortega) Müll. Arg. bark fractions, subfractions, and pure compounds was performed with a murine macrophage cell model RAW 264.7 (Tib-71TM ATCC). The inflammatory process was induced using lipopolysaccharides produced by *Escherichia coli* (LPS), applying the fractions, subfractions, and pure compounds as inhibitory treatments of proinflammatory cytokines. The stages of the evaluation are detailed below.

#### 4.5.1. Cell Culture and Cytotoxicity Assay

A murine macrophage cell line RAW 264.7 was cultured in Dulbecco’s modified Eagle’s medium in an F-12 nutrient mixture (DMEM/F12 medium) supplemented with 10% heat-inactivated fetal bovine serum (FBS). Cells were maintained in a humidified atmosphere containing 5% CO_2_ at 37 °C and subcultured by scraping and seeding in 25 cm^2^ flasks. To assess cell viability, 2 × 10^4^ cells/well in 200 μL of medium were seeded into a 96-well plate and incubated for 24 h.

Subsequently, cells were treated with fractions, subfractions, and pure compounds at various concentrations (3.125, 6.25, 12.5, 25, and 50 μg/mL for fractions and subfractions; 3.75, 7.5, 15, 22.5, and 30 μg/mL for pure compounds) using DMSO as vehicle (0.21%, *v*/*v*), indomethacin (30 μg/mL) as positive control, and untreated cells as negative control. After 2 h, inflammation was induced with lipopolysaccharide (LPS) at a concentration of 4 μg/mL (for wells with extracts, vehicle, indomethacin, and 100% stimulus control) as a proinflammatory stimulus and without LPS (negative control), incubating for 22 h.

Cell viability was determined using the 3-(4,5-dimethylthiazol-2-yl)-5-(3-carboxymethoxyphenyl)-2-(4-sulfophenyl)-2*H*-tetrazolium (MTS) assay, by adding 20 μL of MTS solution to each well, before incubating for another 4 h. Optical density was measured at 490 nm in a microplate reader.

Percent cell viability (%CV) was calculated using the following equation:$$\%CV=\left(\frac{a_{S}}{{\bar{a}}_{LPS}}\right)\times 100$$
where a_S_ = the absorbance of the sample and ${\bar{a}}_{\mathrm{LPS}}$ = the average LPS absorbance.

#### 4.5.2. Nitric Oxide (NO) Inhibition

After cell viability determination, the cell-free supernatants were collected and used for nitric oxide (NO) quantification. Nitrite ion (NO_2_^−^), the stable final product of NO, is an indicator of NO production in cell-free supernatants and was measured according to the Griess reaction. A volume of 50 μL of supernatant from each extract was mixed with 100 μL of Griess reagent (50 μL of 1% sulfanilamide and 50 μL of 0.1% *N*-(1-naphtyl) ethylenediamine dihydrochloride in 2.5% phosphoric acid) for 10 min at room temperature. The optical density of the mixture, at 540 nm (OD_540_), was measured with a microplate reader and the concentration of nitrite in the samples prepared in fresh culture medium was calculated using a NaNO_2_ standard curve [73,74].

The percentage inhibition of nitric oxide was determined using the following steps:

(1)Using the concentrations 0, 1, 5, 10, 10, 20, 40, 60, 60, and 100 µg/mL of NaNO_2_, a calibration curve was determined.
$$a=0.0075\times c_{Na{NO}_{2}}-0.0086$$
where *a* = absorbance and $c_{Na{NO}_{2}}$ = the concentration of sodium nitrate.

(2)The corrected absorbance, (*a_c_*), was calculated for each fraction and subfraction at concentrations of 0, 3.125, 6.25, 12.5, 25, and 50 µg/mL, and for pure compounds at concentrations of 0, 3.75, 7.5, 15, 22.5, and 30 µg/mL, using the difference
$$a_{c}=a_{S}-{\bar{a}}_{NaNO_{2}}^{0}$$
where *a_s_* = the absorbance of the sample and ${\bar{a}}_{NaNO_{2}}^{0}$ = the average absorbance at zero concentration of the NaNO_2_ curve.

(3)The concentration of *NaNO*_2_ (µM) present in each of the fractions, subfractions, and pure compounds was determined using the following equation:
$$c_{Na{NO}_{2}}^{\mu M}=\frac{a_{c}+c_{Na{NO}_{2}}}{a_{S}}$$
where $c_{Na{NO}_{2}}^{\mu M}$ = the micromolar concentration of sodium nitrate.

(4)The percentage of NaNO_2_ in each fraction, subfraction, and pure compound was obtained using the following equation:
$$\%Na{NO}_{2}=\left(\frac{c_{Na{NO}_{2}}^{\mu M}}{{LPS}_{Na{NO}_{2}}^{\mu M}}\right)\times 100$$
where ${LPS}_{Na{NO}_{2}}^{\mu M}$ = the maximum micromolar concentration of sodium nitrate.

(5)Finally, the percentage inhibition of nitric oxide (%I_NO_) was calculated using the following equation:


$$\%I_{NO}=100-\%Na{NO}_{2}$$


#### 4.5.3. Statistical Analysis

The effects of fractions, subfractions, and compounds at different concentrations on cell viability and NO inhibition were studied using two factor completely randomized designs. Experimental units were the cells, which were cultured as described in Section 4.5.1, and three independent replicates of each treatment combination were run. Two-way ANOVA and Tukey’s comparison tests were used for data interpretation, and mean profile graphs were constructed to assess patterns among treatments. In addition, *p*-values < 0.05 were considered statistically significant. The statistical package MINITAB 19 was used.

## 5. Conclusions

By subsequent fractionation, using column chromatography (CC), it was possible to purify and identify the compounds campesteryl palmitate, n-heptyl ferulate, hexadecanoic acid, methyl ester, and a mixture of the compounds brassicasterol, campesterol, β-sitosterol, and stigmasterol.

The compounds and the mixture of free and esterified sterols did not show toxicity in RAW 264.7 murine macrophage cells at any of the studied concentrations.

The compound campesteryl palmitate, an esterified sterol, did not show anti-inflammatory activity in RAW 264.7 murine macrophage cells.

The aromatic compound n-heptyl ferulate showed anti-inflammatory activity in RAW 264.7 murine macrophage cells at 30 μg/mL.

The mixture of sterols (brassicasterol, campesterol, β-sitosterol, and stigmasterol) exhibited anti-inflammatory activity at 22.5 μg/mL and above, which we attribute to the presence of free and esterified sterols.

We conclude that the ethyl acetate extract obtained from *Jatropha cordata* (Ortega) Müll. Arg. bark is a potential source of bioactive compounds with significant anti-inflammatory activity.

## 6. Recommendations

In the future, studies should focus on evaluating the anti-inflammatory activity of the compound n-heptyl ferulate and the mixture of free and esterified sterols (brassicasterol, campesterol, β-sitosterol, and stigmasterol) in animal models to determine whether they maintain their anti-inflammatory effect and do not exhibit toxicity.

Using the supernatant obtained from the cell viability assessment, we suggest applying ELISA kits to evaluate pro- and anti-inflammatory cytokines.

In addition, an in silico study of the compounds campesteryl palmitate and n-heptyl ferulate and the mixture of sterols and triterpenes should be performed to determine whether they present other biological activities of interest.

## Figures and Tables

**Figure 1 plants-12-03780-f001:**
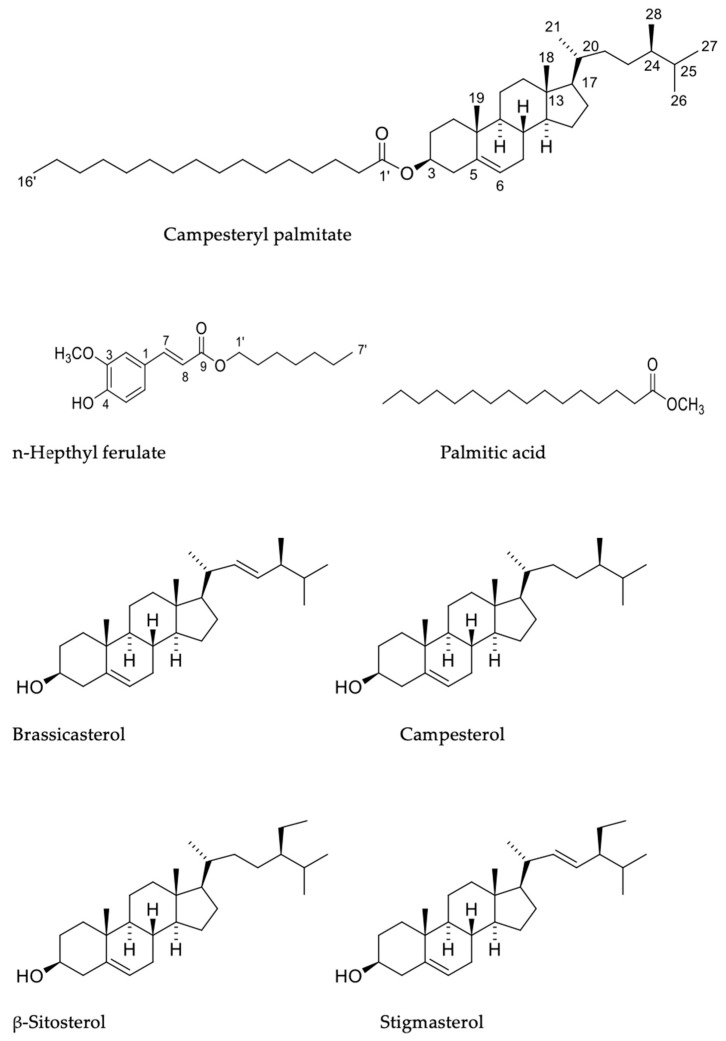
Structure of the identified compounds from *Jatropha cordata* (Ortega) Müll. Arg. bark: campesteryl palmitate, n-heptyl ferulate, brassicasterol, campesterol, β-sitosterol, and stigmasterol.

**Figure 2 plants-12-03780-f002:**
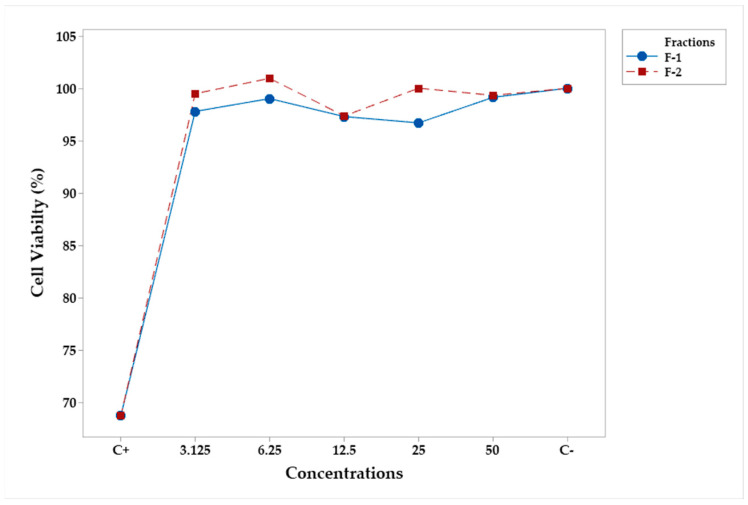
Mean percentage profiles of cell viability in RAW 264.7 murine macrophages, stimulated with LPS and treated with fractions of various concentrations.

**Figure 3 plants-12-03780-f003:**
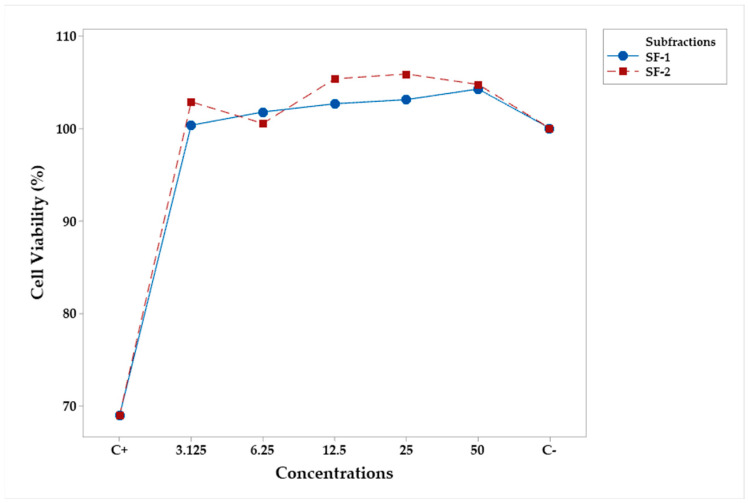
Mean percentage profiles of cell viability in RAW 264.7 murine macrophages, stimulated with LPS and treated with subfractions of various concentrations.

**Figure 4 plants-12-03780-f004:**
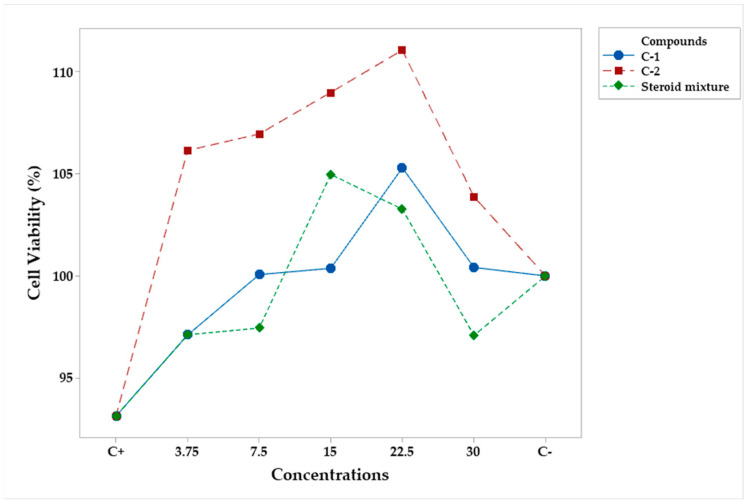
Mean percentage profiles of cell viability in RAW 264.7 murine macrophages, stimulated with LPS and treated with pure compounds and the steroid mixture at various concentrations. LPS and LPS + Indo (Indomethacin) were the positive (C+) and the negative (C−) controls, respectively.

**Figure 5 plants-12-03780-f005:**
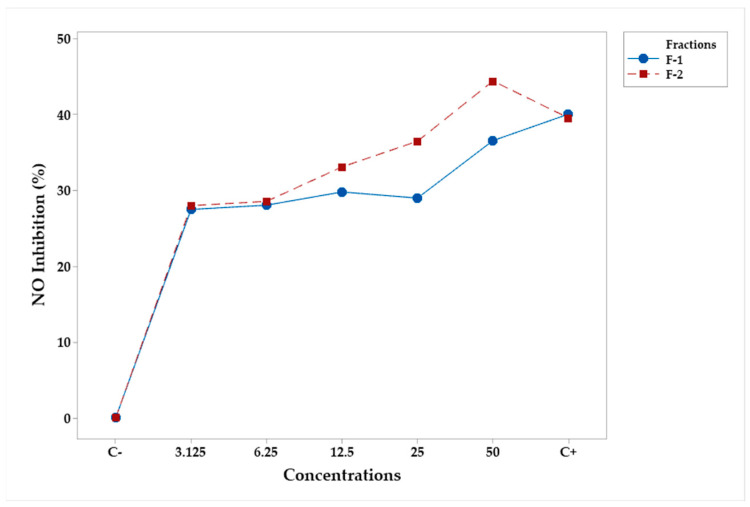
Mean percentage profiles of NO inhibition in RAW 264.7 murine macrophages, stimulated with LPS and treated with fractions. LPS and Indo (Indomethacin) were the positive (C−) and the negative (C+) controls, respectively.

**Figure 6 plants-12-03780-f006:**
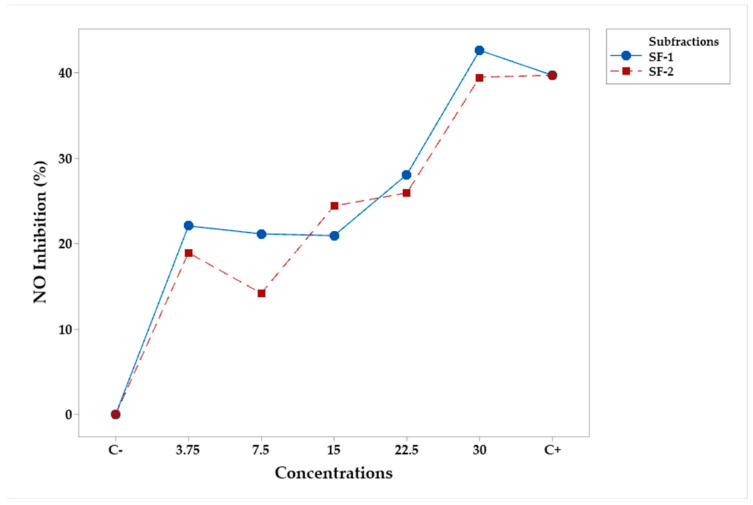
Mean percentage profiles of NO inhibition in RAW 264.7 murine macrophages, stimulated with LPS and treated with subfractions at various concentrations. LPS and Indo (Indomethacin) were the positive (C−) and the negative (C+) controls, respectively.

**Figure 7 plants-12-03780-f007:**
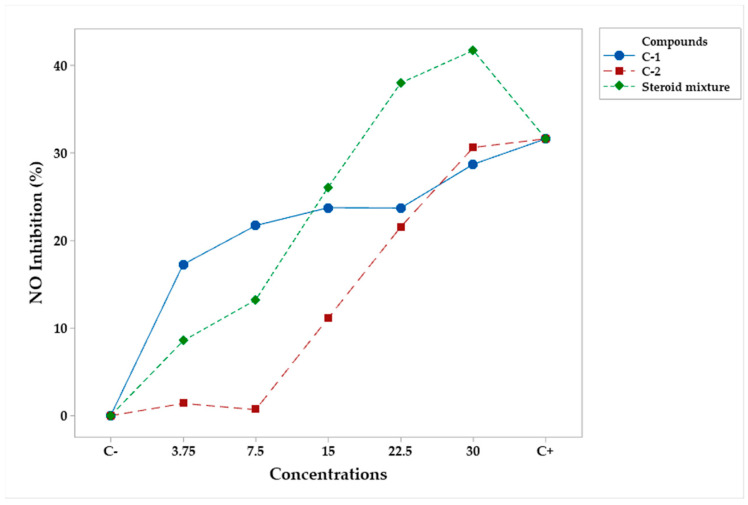
Mean percentage profiles of NO inhibition in RAW 264.7 murine macrophages, stimulated with LPS and treated with pure compounds and the mixture at various concentrations. LPS and Indo (Indomethacin) were the positive (C−) and the negative (C+) controls, respectively.

**Table 1 plants-12-03780-t001:** ANOVA of fractions and concentrations for cell viability.

Source	DF	Contribution	SS	MS	F	*p*-Value
Fractions	1	0.21%	22.15	22.15	1.81	0.18
Concentrations	6	91.24%	9462.2	1577.0	129.1	0.00
Fraction × Concentrations	6	0.30%	30.73	5.12	0.42	0.86
Error	70	8.25%	855.40	12.22		
Total	83	100.00%				

*p*-value < 0.05 was considered statistically significant.

**Table 2 plants-12-03780-t002:** Comparisons of cell viability means for concentrations using Tukey’s test.

Concentrations	N	Mean (%)	Grouping
C−	12	100.00	A
6.25	12	99.98	A
50	12	99.23	A
3.125	12	98.64	A
25	12	98.35	A
12.5	12	97.33	A
C+	12	68.69	B

Means that do not share a letter are significantly different (*p* < 0.05).

**Table 3 plants-12-03780-t003:** ANOVA of subfractions and concentrations for cell viability.

Source	DF	Contribution	SS	MS	F	*p*-Value
Subfractions	1	0.17%	22.5	22.5	1.66	0.20
Concentrations	6	92.1%	11,888.9	1981.5	146.2	0.00
Subfraction × Concentrations	6	0.36%	46.7	7.8	0.57	0.75
Error	70	7.35%	948.7	13.6		
Total	83	100.0%				

*p*-value < 0.05 was considered statistically significant.

**Table 4 plants-12-03780-t004:** Comparisons of cell viability means for concentrations using Tukey’s test.

Concentrations	N	Mean (%)	Grouping
50	12	104.48	A
25	12	104.47	A
12.5	12	103.99	A
3.125	12	101.59	A
6.25	12	101.14	A
C−	12	100.00	A
C+	12	68.94	B

Means that do not share a letter are significantly different (*p* < 0.05).

**Table 5 plants-12-03780-t005:** ANOVA of compounds and concentrations for cell viability.

Source	DF	Contribution	SS	MS	F	*p*-Value
Compounds	2	10.97%	394.5	197.24	7.10	0.00
Concentrations	6	33.04%	1178.0	196.33	7.07	0.00
Compound × Concentrations	12	7.34%	264.1	22.01	0.79	0.66
Error	61	48.65%	1750.2	27.78		
Total	81	100.00%				

*p*-value < 0.05 was considered statistically significant.

**Table 6 plants-12-03780-t006:** Comparisons of cell viability means for compounds using Tukey’s test.

Compounds	N	Mean (%)	Grouping
N-heptyl ferulate	20	104.307	A
Campesteryl palmitate	20	99.483	B
Steroid mixture	42	99.002	B

Means that do not share a letter are significantly different (*p* < 0.05).

**Table 7 plants-12-03780-t007:** Comparisons of cell viability means for concentrations using Tukey’s Test.

Concentrations	N	Mean	Grouping
22.5	12	106.540	A
15	12	104.776	A
7.5	12	101.486	A
30	12	100.458	A
3.75	12	100.121	A
C−	12	100.000	A
C+	10	93.135	B

Means that do not share a letter are significantly different (*p* < 0.05).

**Table 8 plants-12-03780-t008:** ANOVA of fractions and concentrations for NO inhibition.

Source	DF	Contribution	SS	MS	F	*p*-Value
Fractions	1	0.77%	156.8	156.85	1.61	0.208
Concentrations	6	64.92%	13,307.1	2217.85	22.82	0.000
Fraction × Concentrations	6	1.12%	230.4	38.40	0.40	0.880
Error	70	33.19%	6802.8	97.18		
Total	83	100.00%				

*p*-value < 0.05 was considered statistically significant.

**Table 9 plants-12-03780-t009:** Comparisons of NO inhibition means for concentrations using Tukey’s test.

Concentrations	N	Mean	Grouping
50	12	40.4534	A
C+	12	39.7680	A B
25	12	32.7118	A B
12.5	12	31.4245	A B
6.25	12	28.2891	A B
3.125	12	27.7273	B
C−	12	00.0000	C

Means that do not share a letter are significantly different (*p* < 0.05).

**Table 10 plants-12-03780-t010:** ANOVA of subfractions and concentrations for NO inhibition.

Source	DF	Contribution	SS	MS	F	*p*-Value
Subfractions	1	0.20%	60.7	60.71	0.26	0.610
Concentrations	6	46.18%	14,100.5	2350.08	10.17	0.000
Subfraction × Concentrations	6	0.63%	193.7	32.28	0.14	0.990
Error	70	52.99%	16,180.6	231.15		
Total	83	100.00%				

*p*-value < 0.05 was considered statistically significant.

**Table 11 plants-12-03780-t011:** Comparisons of NO Inhibition Means for Concentrations using Tukey’s test.

Concentrations	N	Mean	Grouping
30	12	41.0228	A
C+	12	39.6801	A
22.5	12	26.9936	A B
15	12	22.6562	A B
3.75	12	20.4849	B
7.5	12	17.6513	B C
C−	12	00.0000	C

Means that do not share a letter are significantly different.

**Table 12 plants-12-03780-t012:** ANOVA for compounds and mixture NO inhibition.

Source	DF	Contribution	SS	MS	F	*p*-Value
Compounds	2	5.06%	957.7	487.8	3.76	0.029
Concentrations	6	48.28%	10,389.4	1731.6	13.35	0.000
Compound × Concentrations	12	8.19%	1721.3	143.4	1.11	0.371
Error	63	38.89%	8171.2	129.7		
Total	83	100.00%				

*p*-value < 0.05 was considered statistically significant.

**Table 13 plants-12-03780-t013:** Compound mean comparisons using Tukey’s test.

Compounds	N	Mean	Grouping
Steroid mixture	21	22.7233	A
Campesteryl palmitate	42	20.9525	A
N-heptyl ferulate	21	13.8513	B

Means that do not share a letter are significantly different.

**Table 14 plants-12-03780-t014:** Compound mean comparisons using Tukey’s test.

Concentrations	N	Mean	Grouping
30	12	33.6608	A
C+	12	31.5935	A
22.5	12	27.7326	A
15	12	20.3163	A B
7.5	12	11.8629	B C
3.75	12	9.0637	B C
C−	12	0.0000	C

Means that do not share a letter are significantly different.

## Data Availability

Not applicable.

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
