# Peer review of "Bioactivity of Fractions and Pure Compounds from Jatropha cordata (Ortega) Müll. Arg. Bark Extracts"

_plants, 2023, doi:10.3390/plants12213780_

Round 1

Reviewer 1 Report

Comments and Suggestions for Authors

The manuscript entitled 'Bioactivity of Fractions and Pure Compounds from Jatropha cordata Bark Extracts' presents comprehensive studies of the bark of a medicinal plant, which are performed at the appropriate level.

The main notes:

1.      The moderate check is required for the English language and style through the text. For instance, in еhe Abstract (lines 27-30): 'Drugs for chronic inflammation can cause gastric ulcers, and hepatic and renal issues. An alternative for chronic inflammation is natural bioactive compounds, which present low side effects…' Perhaps these sentences should be rephrased, for instance:  'Medicines of synthetic origin, which are widely used in the treatment of chronic inflammation, can cause gastric ulcers as well as hepatic and renal issues. A good alternative for the treatment of chronic inflammatory processes can be the development and use of medicines from herbal raw material…. '.

2.      I suggest writing the names of molecules in lowercase and without using italics (in the Abstract as well as in the text).

3.      In the introduction, it is worth indicating to which botanical family the plant belongs – Euphorbiaceae.

4.      After the double name of the plant, at the first mention, the abbreviated surnames of the scientists who discovered this species for science should be indicated: Jatropha cordata (Ortega) Müll. Arg.  (according to 'Plants of the World Online | Kew Science' https://powo.science.kew.org/taxon/urn:lsid:ipni.org:names:350198-1). It is worth also to indicate that is Jatropha cordata native to Mexico (Central etc).

5.      It needs to improve the aim and rationale of the study in the end of the Introduction section (lines 55-62).

6.      Line 339 (4. Materials and Methods) - It is necessary to briefly describe how raw materials were collected and dried; also note whether it is from wild or cultivated plants.

7.      Conclusions should be formulated in the form of 1-2 paragraphs, no more. It is worth making them more concise in order to concisely reflect the main achievements of the research.

8.      The list of references regarding Jatropha species could be supplemented. There is a lot of such data in the PubMed database, for example:

https://pubmed.ncbi.nlm.nih.gov/?term=Jatropha+2023&sort=pubdate&size=20

Comments on the Quality of English Language

Moderate editing of English language required

Author Response

Dear revisor 1

First, we greatly appreciate your pertinent recommendations to improve the quality of our

article submi:ed for publication in the journal Plants.

We have followed, one by one, all your critics, comments, and some suggestions to complete

a scientific paper with sufficient merit for publication. In particular,

(1) We found help from a researcher with strong background in writing technical

English, who kindly read and made grammar and style corrections in our writing.

(2) Abstract and Introduction have included your recommendations to correct problems

of word choice in the previous version.

(3) The changes requested by you have all been incorporated as complete as possible,

without substantially changing the meaning of the ideas in the text.

(4) The origin of J. cordata is specifically established, the source of the raw material used

in this study is explicitly given, the rationale of the study has been clarified and the

conclusions have been improved without exaggerate our findings.

(5) Some references concerning related research work on other species of Jatropha have

been incorporated.

Reviewer 2 Report

Comments and Suggestions for Authors

Comments,

Compound names should not be in italic form

To start the review process, I have some concerns about this article

What is different between the submitted article and the published article with the same authors?

Plants (Basel). 2023 Jan 26;12(3):560.

doi: 10.3390/plants12030560.

Phytochemical Characterization and In Vitro Anti-Inflammatory Evaluation in RAW 264.7 Cells of Jatropha cordata Bark Extracts

Comments on the Quality of English Language

Minor

Author Response

Dear revisor 2

First, we greatly appreciate your pertinent recommendations to improve the quality of our

article submi:ed for publication in the journal Plants.

(1) Your main recommendation consisted of improving our English grammar and Style.

We found help from a researcher with strong background in writing technical

English, who kindly read and made corrections in our writing.

Reviewer 3 Report

Comments and Suggestions for Authors

The MS entitled ``Bioactivity of Fractions and Pure Compounds from Jatropha cordata Bark Extracts`` by Jiménez-Nevárez et al. reported the evaluation of Jatropha cordata extracts their cytotoxicity and anti-inflammatory activity. Additionally, sterols and sterol mixture were obtained by column chromatography and evaluated for anti-inflammatory and cytotoxic activities were evaluated using RAW 264.7 murine macrophage cells. The MS cannot be accepted for publication.

Major issues

The current work is a continuation of the previously published wok by them in the same Plants Journal (Jiménez-Nevárez, Y. B., Angulo-Escalante, M. A., Montes-Avila, J., Guerrero-Alonso, A., Christen, J. G., Hurtado-Díaz, I., Heredia, J. B., Quintana-Obregón, E. A., & Alvarez, L. (2023). Phytochemical Characterization and In Vitro Anti-Inflammatory Evaluation in RAW 264.7 Cells of Jatropha cordata Bark Extracts. Plants (Basel, Switzerland), 12(3), 560. https://doi.org/10.3390/plants12030560) that proved the anti-inflammatory activity of the tested plant and its fraction. The current MS lacks novelty and the obtained findings in this work have no significant scientific contribution.  Since, the identified compounds in this work are well-known compounds regarding their structural characterization and anti-inflammatory activity.

Authors should discuss why they carried out such activity testing for these compounds, although the majority of them had been tested previously for anti-inflammatory activity.

Structural characterization and NMR part for all compounds should be removed (Section 2.2 RMN and section 4.3. RMN) or moved to the discussion.

All spectral (GCMS and NMR (1D and 2D NMR)) data should be provided as supplementary.

Introduction is weak and needs more about the folk uses and biological activities of the Jatropha cordata.

I could not find the results of LPS = Lipopolysaccharides and Indo = Indomethacin in figure 2. Also, the control results in figure 3. In both figures 2 and 3 all abbreviations and symbols should be clarified in the legends. The figures are confusing.

Authors should provide a % similarity report using iThenticate Program.

Minor issues

Extensive English editing is needed. There are sentences without verbs. What is ``RMN``?.

The compounds names should not be italicized.

Full name of all abbreviations should be mentioned.

Comments on the Quality of English Language

Extensive English editing is needed.

Author Response

Dear reviewer 3

It is true on your part that this article exhibits work and results closely related to a previous

article by the same authors. It is necessary to argue from our part, that the first article scope

only included the analysis of crude extracts of J. cordata extracts using methanol, ethyl

acetate, and hexane as solvents. It is well known in the published literature on this theme

that it is difficult to identify the specific pure compounds responsible for the observed

bioactivities of the extracts, so our group considered pertinent to continue fractionation the

most promising extract until reaching the pure compounds or mixture of compounds

responsible for such bioactivities. This additional information could be useful in further

development of more effective products for the treatment of health conditions and some

other chronic diseases. Of course, it remains to continue with the analysis of the other

extracts in search for compounds with bioactivities on other diseases. We believe that once

some new compounds may be found, new medicines and therapies might be developed

with further research.

We have also absolutely clear that our compounds found have been already reported in the

literature, however, some complete structures that we found of these compounds were not

known and this provides an opportunity for further and deeply studies on these compounds

that can provide a beBer understanding of their mode of action, and thus, obtain beBer

medicine to treat some chronic problems and help our patients.

We have simplified the section concerning the structural characterization and antiinflamatory

activity and provided more details on the structural characterization by NMR

in a appendix.

The arguments about cultural uses by the population of some parts of J. cordata have been

extended as well as its biological activities. The results from LPS and Indo have been

explicitly introduced in the statistical analysis of results, while explaining their roll as

positive and negative control for the different concentrations of fractions, subfractions and

compounds tested experimentally and analyzed by analysis of variance.

The Abbreviations used have been unified in all cases to avoid confusion in the analysis,

interpretation, and discussion of results.

Legends in each figure have been clarified to clearly convey the meaning of the different

graphs and to facilitate its interpretation.

Finally, English grammar and style was review by an experienced research on these topics,

and we hope the English quality now might deserve recommendation for the publication or

our work.

Reviewer 4 Report

Comments and Suggestions for Authors

Dear Authors:

In your manuscript, Bioactivity of Fractions and Pure Compounds from Jatropha cordata Bark Extracts, the bioactivity of several naturally occurring compounds isolated from the bark of J. cordata was examined regarding their toxicity and anti-inflammatory activity as evidenced by their NO inhibition ability.  The results are very promising and clearly presented, showing the potential in future study that can be focused on the bioactivity of the natural bioactive compounds that act as an alternative for inflammation.  I personally do not have major objections regarding the publication of the manuscript except a few typos that need to be corrected prior to its resubmission to the journal, Plants, and they are listed below.  In addition, based on your findings and previous results from other groups, esterification of certain naturally occurring biochemicals appears to exhibit certain biological activities, such as what occurs in Compound 2, n-Heptyl ferulate, in this study.  From a structural point of view, can you briefly explain how this happens?  The enhanced bioactivity of naturally occuring compounds due to their esterification may help us explore more in future study, e.g., drug design.

In lines 76, and 383, it should be: NMR, NOT RMN.

In line 156, for the structure of palmitic acid, the ester group, −CO2CH3 should be replaced by a carboxylic group, −CO2H.

In line 453, the term, %CV, should not be in a superscript format.

In line 458, the term, āLPS, should not be in a superscript format.

In line 460, nitrite ion should be typed like N.  Or, please download the WORD file (Name: Nitrite_1) attached along with this comment to copy the chemical formula of the ion, nitrite.

In lines 475, 491, and 498, it should be: nitrite, NOT nitrate.

Author Response

Reviewer 4

First, we greatly appreciate your pertinent recommendations to improve the quality of our

article submi:ed for publication in the journal Plants.

He made no comments or suggestions for improvement. The changes in notation have been

completely a:ended.

(1) Your main recommendation consisted of improving our English grammar and Style.

We found help from a researcher with strong background in writing technical

English, who kindly read and made corrections in our writing.

Reviewer 5 Report

Comments and Suggestions for Authors

The article aimed at demonstrating the in vitro anti-inflammatory activity of purified fractions from Jatropha cordata. The subject is of great interest in a biological as well as pharmacologic interest. The in vitro approach is necessary for the selection of compound although the link between these types of results and the final in vivo activity remain elusive in most of the case.

Introduction:

Figure 2, Figure 3, and Figure 4

Legends must be adapted to the figure (example in fig 1: C- and C+ and LPS INDO?) add µM to the Concentrations axis.

Please add precision about the statistical analysis and number of experiments and number of measured points for each experiment in each Figure

Conclusion

This very interesting part is unfortunately focused on many biological activities unrelated to the current objective of the article. I suggest eliminating large parts of the conclusion and perhaps giving more precision on the current results.

Line 259 « Compound 2 displayed an important anti-inflammatory effect » please add a statistical analysis.

Bibliography on palmitic acid seems not necessary as it is a well-known compound.

Material and methods

Purification steps and characterization are too far away from my current knowledge!!!

For cellular experiments:

Please precise the nature of the LPS as cell stimulation is deeply related to this stimulant.

Indomethacin is a good control as anti-inflammatory agent without strong cytotoxic affect. Have you tried a cytotoxic compound in your experiment to demonstrate that your experimental conditions are able to demonstrate cytotoxicity?

Perspectives:

Please add complementary in vitro experiments as the current results may be not sufficient for further in vivo expensive experiments.

Author Response

Reviewer 5

English grammar and style have been improved for a be5er presentation of the contents I

the article,

Figures and legends have been properly corrected for a easier identification of each case.

Unfortunately, even though any product presented cell toxicity, only few of them show

antiinflamatory activity at high concentrations.

Legends have been corrected in Figures 2, 3 and 4. ?M units were adapted to the

concentration axis.

The statistical analysis have been completely rewri5en using ANOVa analysis, Tukey

comparisons tests for mean differences and profile graphs to complement the interpretation.

Conclusions have focused in the antiinflamatory properties ot the compounds found.

The rol of LPS and LPS + Indo as a control treatment have been clarified.

A recommendation has been made to consider cytokines assessments by ELISA kits.

Round 2

Reviewer 2 Report

Comments and Suggestions for Authors

Authors didn't answer my comments,

Compound names should not be in italic form

To start the review process, I have some concerns about this article

What is different between the submitted article and the published article with the same authors?

Plants (Basel). 2023 Jan 26;12(3):560.

doi: 10.3390/plants12030560.

Phytochemical Characterization and In Vitro Anti-Inflammatory Evaluation in RAW 264.7 Cells of Jatropha cordata Bark Extracts

They said that

Author's Notes

Dear revisor 2

First, we greatly appreciate your pertinent recommendations to improve the quality of our

article submitted for publication in the journal Plants.

(1) Your main recommendation consisted of improving our English grammar and Style.

We found help from a researcher with strong background in writing technical

English, who kindly read and made corrections in our writing.

Comments on the Quality of English Language

Moderate changes should be done

Author Response

Reviewer 2:

Comment 1. Compound names should not be in italic form

Answer:

The changes requested by you have all been incorporated as complete as possible, without substantially changing the meaning of the ideas in the text.

Comment 2.

What is different between the submitted article and the published article with the same authors?

Answer:

It is trueonyour part that this article exhibits work and results closely related toa previous article bythe same authors. Itis necessaryto arguefrom our part,that the first article scope only included the analysis of crude extracts of J. cordataextracts using methanol, ethyl acetate, and hexane as solvents. It is well known in the published literature on this theme that it is difficult to identify the specific pure compounds responsible for the observed bioactivities of the extracts, so our group considered pertinent to continue fractionation the most promisingextract until reaching the pure compounds or mixture of compounds responsible for such bioactivities. This additional information could be useful in further development of more effective products for the treatment of health conditions and some other chronic diseases. Of course, it remains to continue with the analysis of the other extracts in search for compounds with bioactivities on other diseases. We believe that once some new compounds may be found, new medicines and therapies might be developed with further research.

Reviewer 3 Report

Comments and Suggestions for Authors

Authors clarified the needed points and did the suggested correction.

Author Response

Dear reviewer3

It is trueonyour part that this article exhibits work and results closely related toa previous article bythe same authors. Itis necessaryto arguefrom our part,that the first article scope only included the analysis of crude extracts of J. cordataextracts using methanol, ethyl acetate, and hexane as solvents. It is well known in the published literature on this theme that it is difficult to identify the specific pure compounds responsible for the observed bioactivities of the extracts, so our group considered pertinent to continue fractionation the most promisingextract until reaching the pure compounds or mixture of compounds responsible for such bioactivities. This additional information could be useful in further development of more effective products for the treatment of health conditions and some other chronic diseases. Of course, it remains to continue with the analysis of the other extracts in search for compounds with bioactivities on other diseases. We believe that once some new compounds may be found, new medicines and therapies might be developed with further research.

We havealso absolutely clear that our compounds foundhave been already reported in the literature, however, some complete structures that we found of these compounds were not known and this provides an opportunity for further and deeply studies on these compounds that can provide a better understanding of their mode of action, and thus, obtain better medicine to treat some chronic problems and help our patients.

We have simplified the section concerning the structural characterization and antiinflamatory activity and provided more details on the structural characterization by NMR in a appendix.

The arguments about cultural uses by the population of some parts of J. cordatahave been extendedas well as its biological activities. The results from LPS and Indo have been explicitly introduced in the statistical analysis of results, while explaining their roleas positive and negative control for the different concentrations of fractions, subfractions and compounds tested experimentally and analyzed by analysis of variance.

The Abbreviations used have been unified in all cases to avoid confusion in the analysis, interpretation, and discussion of results.

Legends in each figure have been clarified to clearly convey the meaning of the different graphs and to facilitate its interpretation.

Finally, English grammar and style was review by an experienced researchedon these topics, and we hope the English quality now might deserve recommendation for the publication or our work.

Reviewer 5 Report

Comments and Suggestions for Authors

thank for the modifications...

Author Response

Reviewer5

English grammar and style have been improved for a better presentation of the contents I the article,

Figures and legends have been properly corrected for a easier identification of each case.

Unfortunately, even though any product presented cell toxicity, only few of them show antiinflamatory activity at high concentrations.

Legends have been corrected in Figures 2, 3 and 4. mM units were adapted to the concentration axis.

The statistical analysis hasbeen completely rewritten using ANOVAanalysis, Tukey comparisons tests for mean differences and profile graphs to complement the interpretation.

Conclusions have focused in the antiinflamatory properties ofthe compounds found.

The roleof LPS and LPS + Indo as a control treatment have been clarified.

Arecommendationhas been incorporated in the text,atthe Section6on Recommendations,to consider cytokines assessments by ELISA kits.

Round 3

Reviewer 2 Report

Comments and Suggestions for Authors

In my opinion, the article lacks the novelty of overlapping with previous published articles. I can not find new information. Both articles have NMR identification with the same bioactivity.

Comments on the Quality of English Language

Minor corrections.